# EditTrack: Detecting and Attributing AI-assisted Image Editing

## Abstract

In this work, we formulate and study the problem of *image-editing detection* and *attribution*: given a *base image* and a *suspicious image*, detection seeks to determine whether the suspicious image was derived from the base image using an AI editing model, while attribution further identifies the specific editing model responsible. Existing methods for detecting and attributing AI-generated images are insufficient for this problem, as they focus on determining whether an image was AI-generated/edited rather than whether it was edited from a particular base image. To bridge this gap, we propose *EditTrack*, the first framework for this image-editing detection and attribution problem. Building on four key observations about the editing process, EditTrack introduces a novel *re-editing* strategy and leverages carefully designed similarity metrics to determine whether a suspicious image originates from a base image and, if so, by which model. We evaluate EditTrack on five state-of-the-art editing models across six datasets, demonstrating that it consistently achieves accurate detection and attribution, significantly outperforming five baselines.

## 1 Introduction

Recent advances in AI editing models (OpenAI, 2024; Couairon et al., 2023; Deng et al., 2025) enable users to transform existing images into high-quality outputs guided by natural language instructions. While such techniques empower non-expert users to modify images according to their preferences and greatly enhance the diversity of synthetic visual content, their misuse in copyright infringement and deepfake creation raises serious societal concerns. For example, a user might request an editing model to 'turn the dog in the artwork into a cat', producing a cat artwork that preserves the original artistic style while replacing the dog. The user could then claim ownership of this edited image, infringing upon the copyright of the original artist (Akers, 2024). Similarly, a user might prompt the model to 'replace the person in the photo with Donald Trump', thereby generating a deepfake image depicting him in inappropriate contexts or scenes.

At the center of these societal concerns lies the *image-editing detection and attribution problem*: given a *base image* and a *suspicious image*, detection determines whether the suspicious image was derived from the base using an AI editing model, while attribution further identifies the specific model responsible. A tool addressing this problem has broad practical applications. For example, an artist could use it to verify whether a suspicious image (e.g., the cat artwork above) was generated from their original work (e.g., the dog artwork) using AI, and further attribute the edit to a particular model. Similarly, forensic analysts and law enforcement agencies could determine whether a deepfake (e.g., the example above) was created from a specific base image and identify the editing model, thereby aiding cybercrime investigations and tracing back to the criminals.

Existing methods (Sun et al., 2024; Sha et al., 2024) for detecting and attributing AI-generated images are insufficient for our problem. These approaches focus on determining whether an image was generated by AI and, if so, identifying the specific model. While they can be adapted to detect whether a suspicious image has been AI-edited, they cannot establish whether it was derived from a given base image. Furthermore, although attribution methods can in principle be extended to identify the editing model once a suspicious image has been detected as derived from a base image, their effectiveness is limited because they overlook the unique characteristics introduced by the editing process relative to the base image, as our experiments demonstrate.

To bridge this gap, we propose EditTrack, the *first* framework to explicitly address the image-editing detection and attribution problem. Given a base–suspicious image pair, EditTrack first determines whether the suspicious image was derived from the base image using an AI editing model. If so, it further attributes the editing to the responsible model within a set of candidate editing models.

EditTrack is built on four key observations of the AI-assisted editing process: (1) *robustness*, meaning that an editing model produces similar edited images when given the same base image with semantically similar editing prompts; (2) *stability*, meaning that an edited image remains similar if it is edited again by the same model with a semantically similar editing prompt; (3) *variety*, meaning that different editing models produce distinct edited images even when provided with the same base image and editing prompt; and (4) *dissimilarity*, meaning that if a suspicious image is not originally derived from a base image, no editing model will be able to reproduce it from that base image.

Based on these observations, EditTrack evaluates a base–suspicious image pair by testing whether the suspicious image was edited from the base using a candidate model through a *re-editing* procedure. Specifically, we apply the candidate model to re-edit both the base and suspicious images with a prompt that encodes the differences between them. If the suspicious image truly originates from the base via this model, the re-edited images should closely resemble the suspicious image; otherwise, they should diverge. To quantify this similarity, we adopt six complementary metrics across three categories–*structural*, *semantic*, and *pixel-level similarity*–each capturing distinct facets of image correspondence. We then combine these metrics by framing detection and attribution as a multi-class classification task, where the classifier takes the similarity features of a base–suspicious pair as input and outputs a label indicating either the responsible editing model or that the suspicious image was not derived from the base.

We conduct a comprehensive evaluation across five state-of-the-art image editing models and multiple datasets that cover diverse scenarios. While EditTrack is, to the best of our knowledge, the first framework to directly address the image-editing detection and attribution problem, we also adapt several existing techniques for comparison. Our results show that: (1) EditTrack consistently achieves high detection and attribution accuracy across editing models and datasets, and (2) EditTrack significantly outperforms all baselines. We further validate our design through ablation studies on key components of EditTrack.

## 2 RELATED WORK

**AI-generated image detection and attribution:** AI-generated image detection aims to determine whether an image has been created or edited by AI. Existing techniques fall into two categories: *passive* and *proactive*. Passive detection methods (Sun et al., 2024; Sha et al., 2024) typically identify subtle artifacts or statistical irregularities in AI-generated images, such as inconsistent noise patterns, unnatural textures, or distorted anatomical features (e.g., hands and teeth) that AI models often leave behind. Although these methods can be applied to detect AI-edited images, they do not reveal any information about the original base image. Specifically, given a base–suspicious image pair, these methods can only determine whether the suspicious image has been AI-edited, but not whether it was derived from the given base image. While attribution methods could, in principle, be extended to identify the editing model once a suspicious image is confirmed to originate from a base image, their effectiveness is limited. This limitation arises because they do not account for the unique characteristics introduced by the editing process relative to the base image, as demonstrated by our experiments.

Proactive detection methods (Jiang et al., 2024) embed watermarks into AI-generated images, enabling later detection by verifying whether the watermark can be extracted. Such methods can, in principle, be applied to image editing detection: the owner of a base image embeds a watermark, and if the watermark is later recovered from a suspicious image, the image can be flagged as edited from the base. However, image watermarks are not robust to AI editing (Zhao et al., 2024). In practice, the watermark embedded in the base image often becomes undetectable after editing, as confirmed by our experiments with the state-of-the-art WAM method (Sander et al., 2025), leading to unreliable detection. Moreover, watermark-based attribution (Jiang et al., 2024) would require each editing model to embed a distinct watermark into the images it generates or edits. This approach depends on cooperation among model providers, which is generally impractical.

**Preventing AI-assisted image editing:** To mitigate the misuse of AI-assisted image editing, one line of defense focuses on preventing editing models from successfully editing a base image. These approaches (Salman et al., 2023; Choi et al., 2025; Wan et al., 2024) add carefully crafted, human-imperceptible perturbations to base images so that, when edited, the perturbations disrupt the process and produce low-quality or unusable edited images. However, such perturbations can be easily removed by adaptive techniques (Nie et al., 2022; Sandoval-Segura et al., 2023; Xue & Chen, 2024), leaving editing still feasible. Our work is complementary to this direction: instead of preventing edits in advance, we target scenarios where edited images have already been produced. By enabling reliable detection and attribution, our method provides a post-hoc mechanism to trace AI-assisted image editing and mitigate its harms.

## 3 PROBLEM FORMULATION

**AI-assisted image editing:** Image editing generally refers to modifying an image–referred to as the *base image* $I_b$–to meet a user's needs. With recent advances in AI, this process is increasingly automated by editing models. Specifically, an editing model $\mathcal{M}$ takes a base image $I_b$ and an *editing prompt* $p_e$ as input, where $p_e$ specifies the desired modifications to $I_b$, and produces an *edited image* $I_e$ that reflects the requested changes. Formally, we define AI-assisted image editing as follows:

**Definition 1** (AI-assisted Image Editing). Given a base image $I_b$ and an editing prompt $p_e$, an editing model $\mathcal{M}$ produces an edited image $I_e$ that reflects the requested changes, i.e., $I_e = \mathcal{M}(I_b, p_e)$.

Note that the form of the editing prompt $p_e$ may vary across editing models. For example, DiffEdit (Couairon et al., 2023) requires both a description of the base image and a description of the desired edited image as the editing prompt, which it then uses to generate a mask that highlights the regions of the base image to be modified. In contrast, Step1X-Edit (Liu et al., 2025) does not require a description of the base image; instead, it only uses a description of the intended modifications as the editing prompt.

**Image-editing detection and attribution:** Given a base image $I_b$, a suspicious image $I_s$, and a set of candidate editing models $S = \{\mathcal{M}_1, \mathcal{M}_2, \ldots, \mathcal{M}_n\}$, *image-editing detection* aims to determine whether $I_s$ was derived from $I_b$ using any model in $S$. Once detection confirms editing, *image-editing attribution* further seeks to identify the specific model responsible. Formally, $I_s$ is considered an edited version of $I_b$ if an editing model $\mathcal{M}' \in S$ and an editing prompt $p_e'$ can be found such that $I_s = \mathcal{M}'(I_b, p_e')$, and the editing is attributed to $\mathcal{M}'$. The set $S$ is necessary, as attribution is inherently limited to known candidate models. For example, $S$ may consist of widely used open-source or closed-source editing models, though our method is applicable to any such set. Formally, we define the image-editing detection and attribution problems as follows:

**Definition 2** (Image-Editing Detection). Given a base image $I_b$ and a suspicious image $I_s$, image editing detection is to determine whether $I_s$ was derived from $I_b$ using some editing model.

**Definition 3** (Image-Editing Attribution). Given a base image $I_b$, a suspicious image $I_s$, and a set of candidate editing models $S = \{\mathcal{M}_1, \mathcal{M}_2, \ldots, \mathcal{M}_n\}$, image editing attribution is to identify the specific model in $S$ that generated $I_s$ from $I_b$, once $I_s$ has been detected as an edited version of $I_b$.

In this work, we propose EditTrack to address the image-editing detection and attribution problem. EditTrack operates without requiring access to the parameters of candidate editing models, making it applicable to both closed-source and open-source settings. Moreover, it does not rely on the editing prompts used to produce the edited images.

## 4 EDITTRACK

### 4.1 OVERVIEW

**A straightforward but computationally prohibitive solution:** To address the image-editing detection and attribution problem, a straightforward solution based on the above definitions is to search for an editing prompt $p_e'$ for each candidate editing model $\mathcal{M}_i \in S$, where $i = 1, 2, \ldots, n$. If such an editing prompt $p_e'$ can be found that satisfies $I_s = \mathcal{M}_i(I_b, p_e')$, then the suspicious image $I_s$ is detected as edited from the base image $I_b$, and the editing is attributed to model $\mathcal{M}_i$. However, this

approach is impractical due to the immense complexity of the search. Specifically, the prompt space is discrete, the vocabulary is large, and editing prompts can be arbitrarily long, making the search intractable in practice. As a result, even if $I_s$ was indeed edited from $I_b$, the method may fail simply because such an editing prompt cannot be found in practice.

**Our EditTrack:** Instead of searching an editing prompt $p'_e$ like in the above straightforward solution, we make four key observations about AI-assisted image editing, which form the basis of EditTrack. Specifically, given a base image $I_b$ and a suspicious image $I_s$, we first generate an editing prompt $p'_e$ using a captioning model by comparing the differences between $I_s$ and $I_b$. We then apply each candidate editing model in $S$ to $I_b$ and $I_s$ with $p'_e$ as the editing prompt to generate two re-edited images. If $I_s$ was indeed derived from $I_b$ by a particular model $\mathcal{M}_i$, the re-edited images from $\mathcal{M}_i$ should be highly similar to $I_s$, whereas those from other models should be comparatively dissimilar. A follow-up challenge for EditTrack is quantifying the similarity between a re-edited image and a suspicious image. To address this, we select six metrics to measure the similarity between each re-edited image and the suspicious image $I_s$, resulting in $12n$ features for a given base–suspicious image pair, where $n$ is the number of candidate editing models. To integrate these features for detection and attribution, we train an $(n + 1)$-class classifier that takes the $12n$ features of a base-suspicious image pair as input and outputs a label, indicating either a specific candidate editing model or the non-edited case.

## 4.2 EXTRACTING FEATURES

**Four observations:** We make four key observations about AI-assisted image editing, which we empirically validate in our experiments. For clarity, we present them as follows:

**Observation 1** (Robustness). *Suppose a suspicious image $I_s$ is derived from a base image $I_b$ using an editing prompt $p_e$ and an editing model $\mathcal{M}$. Robustness means that if $p_e$ is replaced with a semantically similar prompt $p'_e$, then $\mathcal{M}$ generates an edited image that remains highly similar to $I_s$, i.e.,*

$$I_s = \mathcal{M}(I_b, p_e) \approx \mathcal{M}(I_b, p'_e) \text{ for } p_e \approx p'_e, \tag{1}$$

*where the notation $\approx$ indicates that two images or prompts are similar.*

**Observation 2** (Stability). *Suppose a suspicious image $I_s$ is derived from a base image $I_b$ using an editing prompt $p_e$ and an editing model $\mathcal{M}$. Stability means that if we apply $\mathcal{M}$ again to $I_s$ with a semantically similar prompt $p'_e$, the resulting re-edited image remains highly similar to $I_s$. In other words, once $\mathcal{M}$ generates $I_s$ from $(I_b, p_e)$, the editing process has converged at $I_s$, i.e.,*

$$I_s = \mathcal{M}(I_b, p_e) \approx \mathcal{M}(I_s, p'_e) \text{ for } p_e \approx p'_e. \tag{2}$$

**Observation 3** (Variety). *Given a base image $I_b$ and an editing prompt $p_e$, variety means that different editing models produce edited images that exhibit distinct variations, i.e.,*

$$\mathcal{M}_i(I_b, p_e) \not\approx \mathcal{M}_j(I_b, p_e) \text{ for } i \neq j, \tag{3}$$

*where the notation $\not\approx$ indicates that two images are comparably dissimilar.*

**Observation 4** (Dissimilarity). *Suppose a suspicious image $I_s$ is not derived from a base image $I_b$ by an editing model $\mathcal{M}$. Dissimilarity means that even when $\mathcal{M}$ is given $I_b$ together with an editing prompt $p'_e$ designed to capture the differences between $I_b$ and $I_s$, the resulting edited image remains dissimilar to $I_s$, i.e.,*

$$I_s \not\approx \mathcal{M}(I_b, p'_e). \tag{4}$$

**Producing re-edited images:** Our EditTrack builds on these observations. Given a base-suspicious image pair $(I_b, I_s)$ and a set of $n$ candidate editing models $S = \{\mathcal{M}_1, \mathcal{M}_2, \ldots, \mathcal{M}_n\}$, we generate two re-edited images using each candidate model. Since the true editing prompt $p_e$ that may have produced $I_s$ from $I_b$ is unavailable, we construct a proxy prompt $p'_e$ using a captioning model (e.g., BLIP-2 in our experiments). Specifically, the captioning model generates descriptions $p_b$ and $p_s$ for $I_b$ and $I_s$, respectively, and we form $p'_e$ as: "Do the image editing task; original prompt: $\{p_b\}$, editing prompt: $\{p_s\}$." For each candidate editing model $\mathcal{M}_i$, we produce two re-edited images: one by applying $p'_e$ to the base image $I_b$, yielding $I^i_{rb} = \mathcal{M}_i(I_b, p'_e)$, and the other by applying $p'_e$ to the suspicious image $I_s$, yielding $I^i_{rs} = \mathcal{M}_i(I_s, p'_e)$.

If $I_s$ is not edited from $I_b$ by any model, then according to Observation 4, the re-edited images from all candidate models will be comparatively dissimilar to $I_s$. However, if $I_s$ is indeed edited from $I_b$ by model $\mathcal{M}_i$, i.e., $I_s = \mathcal{M}_i(I_b, p_e)$, then according to Observations 1 and 2, its re-edited images $I_{rb}^i$ and $I_{rs}^i$ should both be highly similar to $I_s$. In contrast, by Observations 1 and 3, re-edited images $I_{rb}^j$ generated by other models $\mathcal{M}_j$ ($j \neq i$) based on $I_b$ are comparatively dissimilar to $I_s$. Specifically, we have $I_{rb}^j = \mathcal{M}_j(I_b, p_e') \approx \mathcal{M}_j(I_b, p_e) \not\approx \mathcal{M}_i(I_b, p_e) = I_s$. In addition, by Observations 2 and 3, re-edited images $I_{rs}^j$ generated by other models $\mathcal{M}_j$ ($j \neq i$) based on $I_s$ are also comparatively dissimilar to $I_s$. Specifically, we have $I_{rs}^j = \mathcal{M}_j(I_s, p_e') \not\approx \mathcal{M}_i(I_s, p_e') = \mathcal{M}_i(\mathcal{M}_i(I_b, p_e), p_e') \approx \mathcal{M}_i(I_b, p_e) = I_s$.

Therefore, if there exists a candidate model $\mathcal{M}_i$ whose re-edited images $I_{rb}^i$ and $I_{rs}^i$ both exhibit strong similarity to the suspicious image $I_s$, while the re-edited images of other models do not, we conclude that $I_s$ was edited from $I_b$ and attribute the editing to $\mathcal{M}_i$.

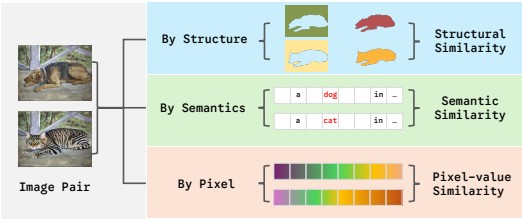

Figure 1: Three similarity categories.

**Extracting image similarity as features:** Another crucial aspect of our method is quantifying the similarity between a re-edited image and the suspicious image. To address this challenge, we adopt six similarity metrics spanning three categories: *structural similarity* (2 metrics), *semantic similarity* (2 metrics), and *pixel-value similarity* (2 metrics), each capturing different facets of similarity. Figure 1 illustrates three categories.

- **Structural similarity:** These metrics focus on the geometric and spatial layout of images, measuring how the overall structure or composition of two images aligns, independent of color, texture, or style. In this category, we consider *structural distance* (Tumanyan et al., 2022) and *pHash* (Zauner, 2010).

- **Semantic similarity:** These metrics capture the high-level conceptual content of images, measuring whether two images depict the same objects, scenes, or ideas, even when their visual appearances differ. In this category, we use the widely adopted *CLIP score* (Radford et al., 2021) and LPIPS (Zhang et al., 2018) to assess semantic similarity.

- **Pixel-value similarity:** These metrics evaluate low-level visual correspondence, focusing on pixels, colors, and textures. They assess whether the basic visual properties of objects are consistent across two images. To do so, we first compute the image histogram for each image, which is a graphical representation that shows the distribution of pixel intensity values, and then quantify similarity using *Intersection score* (Swain & Ballard, 1991) and *Bhattacharyya distance* (Kailath, 2003).

**Validating the four observations:** To empirically validate our observations, we conduct re-editing experiments using two editing models: Step1X-Edit (Liu et al., 2025) and EditAR (Mu et al., 2025). We construct 50 positive base-suspicious pairs $(I_b, I_s)$, where each suspicious image $I_s$ is derived from its corresponding base image $I_b$ using Step1X-Edit with an editing prompt $p_e$, and 100 negative pairs in which $I_s$ is unrelated to $I_b$. We design four groups of re-editing experiments, corresponding to the legends in Figure 2: (1) "Positive, Base, Step1X-Edit": re-editing the base image in each positive pair via Step1X-Edit using our $p_e'$; (2) "Positive, Suspicious, Step1X-Edit": re-editing the suspicious image in each positive pair via Step1X-Edit using $p_e'$; (3) "Positive, Base, EditAR": re-editing the base image in each positive pair via EditAR using $p_e$; and (4) "Negative, Base, Step1X-Edit": re-editing the base image in each negative pair via Step1X-Edit using $p_e'$. For each group, we compute the similarity between every re-edited image and its corresponding suspicious image. Figure 2 presents the distribution of similarity scores across the six metrics.

The results are consistent with our four observations. First, the distributions for "Positive, Base, Step1X-Edit" consistently show high similarity, supporting Observation 1. Second, the distributions for "Positive, Suspicious, Step1X-Edit" also show high similarity, aligning with Observation 2. Third, the distributions for "Positive, Base, EditAR" exhibit lower similarity than the previous two

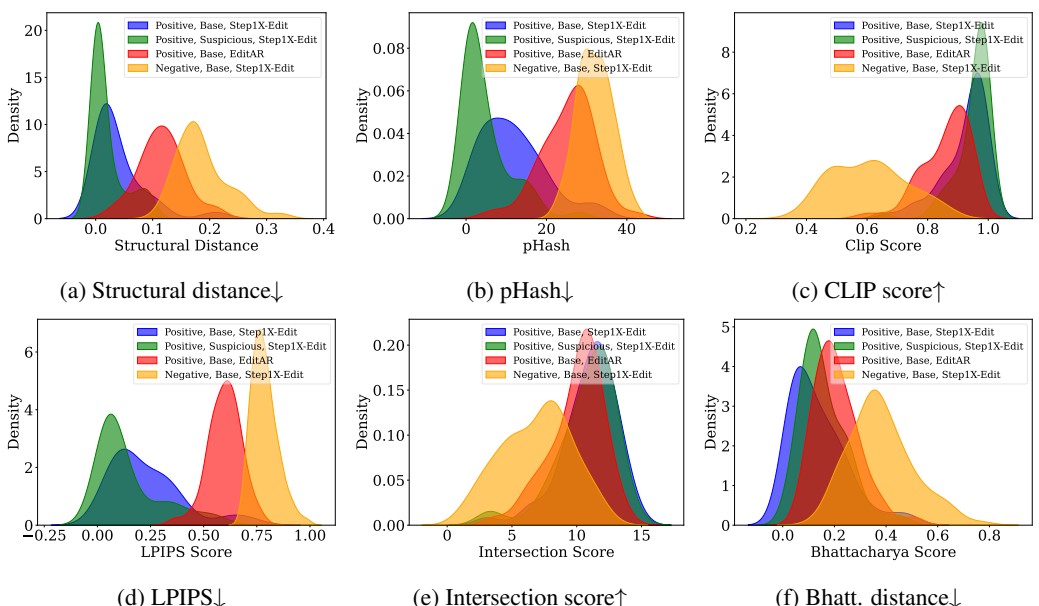

Figure 2: Validation of the four observations. $\uparrow$ / $\downarrow$ indicate that higher / lower values correspond to greater similarity.

cases, validating Observation 3. Finally, the distributions for "Negative, Base, Step1X-Edit" generally show the lowest similarity, consistent with Observation 4.

### 4.3 TRAINING A MULTI-CLASS CLASSIFIER

**Binary classifiers achieve limited performance:** A natural approach to the image-editing detection and attribution problem is to train a *binary classifier* that distinguishes *positive* from *negative* base-suspicious image pairs, which we denote as *EditTrack-Bin*. For each candidate editing model $\mathcal{M}_i \in S$, we collect positive pairs $(I_b, I_s)$ where $I_s$ is derived from $I_b$ via $\mathcal{M}_i$. For each such pair, we generate two re-edited images using a candidate editing model, compute the six similarity metrics between each re-edited image and $I_s$, and aggregate them into a 12-dimensional feature vector labeled as 'positive.' Repeating this process across all $n$ candidate models yields $n$ positive training samples per such pair. Likewise, for negative pairs $(I_b, I_s)$, where $I_s$ is not edited from $I_b$, we extract the same 12 features with each candidate model, producing $n$ negative training samples per pair. These samples are then combined to train the binary classifier.

At test time, given a base-suspicious image pair, we repeat this process with each of the $n$ candidate editing models, producing $n$ 12-dimensional test inputs $x_1, x_2, \ldots, x_n$, where $x_i$ corresponds to $\mathcal{M}_i$. We apply the binary classifier to each $x_i$. If any are classified as positive, the pair is detected as positive, and attribution is made to the model $\mathcal{M}_{i*}$ whose corresponding input $x_{i*}$ has the highest probability of being positive under the classifier.

To capture variety across editing models, we can also train a separate binary classifier for each candidate editing model (denoted as *EditTrack-Bin-Multiple*), aiming to distinguish positive pairs generated by $\mathcal{M}_i$ from negatives. At test time, the $i$-th classifier evaluates the corresponding $x_i$; if any classifier returns positive, attribution is assigned to the model $\mathcal{M}_{i*}$ whose classifier outputs the highest probability of being positive.

However, our experiments show that such binary-classifier approaches yield inaccurate detection and attribution. The core limitation is that they treat editing models in isolation, rather than jointly reasoning over all candidates when making a prediction.

**Training an $(n + 1)$-class classifier:** To overcome the limitations of binary classification, our EditTrack trains an $(n + 1)$-class classifier. For each positive pair $(I_b, I_s)$ generated by candidate

editing model $\mathcal{M}_i$, we apply all $n$ candidate editing models to produce two re-edited images each, yielding $2n$ re-edited images. From these, we compute the six similarity scores between each re-edited image and $I_s$, resulting in a $12n$-dimensional feature vector. We assign label $i$ to this feature vector, indicating that the pair was generated by $\mathcal{M}_i$. Negative pairs are processed in the same way, with their $12n$-dimensional feature vectors assigned label 0. Each training sample therefore consists of a $12n$-dimensional feature vector and a label from $\{0, 1, \ldots, n\}$, which together are used to train the $(n + 1)$-class classifier. At test time, a base-suspicious image pair is represented as a $12n$-dimensional feature vector using the same procedure, and the classifier predicts a label $i^*$. If the predicted label $i^*$ is non-zero, the pair is deemed positive and attributed to the corresponding editing model $\mathcal{M}_{i^*}$.

## 5 EVALUATION

### 5.1 EXPERIMENTAL SETUP

**Editing models:** We evaluate five state-of-the-art image editing models, including four diffusion-based models (DiffEdit (Couairon et al., 2023), FireFlow (Deng et al., 2025), StableFlow (Avrahami et al., 2025), and Step1X-Edit (Liu et al., 2025)) and one autoregressive-based model (EditAR (Mu et al., 2025)). All editing models are implemented using their official repositories and default configurations. All experiments are conducted on a single 80GB A100 GPU.

**Training and testing datasets:** We construct datasets containing both *positive pairs* and *negative pairs* for training and testing. Each pair consists of a base image and a suspicious image. Table 1 summarizes the datasets. We use all training datasets to train the classifier and evaluate performance separately on each testing dataset.

A positive pair means that the suspicious image is generated by editing the base image using an image editing model. For the training dataset, we randomly sample (1) 60 images from Flickr2K (Yang, 2024) and (2) 60 images from WikiArt (Hugging-Face, 2022), yielding 120 base images. For each base image, we manually design an editing prompt, which

Table 1: Summary of our datasets.

| | Dataset | #Train | #Train in total | #Test |
|---|---|---|---|---|
| Positive pairs | Flickr2K | 60 per model | 600 | 50 per model |
| | WikiArt | 60 per model | | 50 per model |
| Negative pairs | MSCOCO | 200 | 600 | 100 |
| | Artwork | 200 | | 100 |
| | Inter4K | 200 | | 100 |
| | Unrelated | - | | 100 |

is then used by the five different image editing models to generate the corresponding suspicious images. Some image pairs and corresponding editing prompts are provided in Figure 4 and 5 in the Appendix. The testing dataset is constructed in the same way, using another 50 base images from Flickr2K and WikiArt, respectively.

A negative pair means that the suspicious image is not edited from the base image. For training, we construct three datasets, in which each base-suspicious image pair depicts similar objects with closely related semantics: (1) images from the same category in MSCOCO (Lin et al., 2014); (2) artwork images with similar styles from Artvee (Artvee, 2024) and the Van Gogh Museum (Gogh, 2024); (3) two frames of a video from Inter4K (Stergiou & Poppe, 2022). Each dataset contains 200 negative base-suspicious pairs. For testing, we construct an additional 100 negative pairs from each dataset, where the suspicious images are not AI-edited. To further evaluate cases in which the suspicious image is AI-edited but unrelated to the base image, we create (4) an *unrelated* dataset consisting of 100 negative pairs. In this dataset, suspicious images are randomly sampled from the testing positive pairs, while base images are randomly sampled from the testing negative pairs.

**Compared methods:** We first construct baselines using state-of-the-art vision-language models (VLMs). Specifically, we prompt QWen2.5-VL (Bai et al., 2025) with a base-suspicious image pair to determine whether the suspicious image is edited from the base image. To enable attribution, we additionally provide an example pair for each candidate editing model and ask QWen to attribute the suspicious image through its in-context learning capability, denoted as *QWen-Prompting*. We also fine-tune QWen on our training datasets and evaluate the fine-tuned model on both detection and attribution tasks, denoted as *QWen-FT*. Similarly, we fine-tune LLaVa-v1.5-7B (HuggingFace,

Table 2: Detection accuracy results across methods and datasets.

| Method | Positive Pairs | | Negative Pairs | | | | Overall Acc. |
|---|---|---|---|---|---|---|---|
| | Flickr2K | WikiArt | MSCOCO | Artwork | Inter4K | Unrelated | |
| QWen-Prompting | 0.084 | 0.108 | 1.000 | 1.000 | 0.930 | 1.000 | 0.490 |
| QWen-FT | 0.052 | 0.040 | 1.000 | 1.000 | 1.000 | 1.000 | 0.470 |
| LLaVa-FT | 0.972 | 0.984 | 0.120 | 0.030 | 0.090 | 0.010 | 0.571 |
| WAM | 0.652 | 0.664 | 0.960 | 1.000 | 0.960 | 0.970 | 0.798 |
| GRE | 0.532 | 0.680 | 0.790 | 0.710 | 0.900 | 0.370 | 0.644 |
| EditTrack-Bin | 0.996 | 0.992 | 0.970 | 0.590 | 0.610 | 1.000 | 0.904 |
| EditTrack-Bin-Multiple | 0.976 | 0.940 | 1.000 | 0.990 | 0.950 | 1.000 | 0.970 |
| EditTrack | 0.984 | 0.972 | 1.000 | 0.980 | 0.980 | 1.000 | 0.983 |

2023), denoted as *LLaVa-FT*. However, since our prompting setup requires 12 input images for attribution, which exceeds LLaVa's maximum token limit, evaluating *LLaVa-Prompting* is not feasible. In addition, we extend the state-of-the-art watermarking method WAM (Sander et al., 2025) as a baseline for the detection task by embedding a watermark into base images and predicting a pair to be positive if the bitwise similarity between the watermark extracted from the suspicious image and the ground-truth watermark exceeds a threshold (0.5 in our experiments). We also extend GRE (Sun et al., 2024), a state-of-the-art method for detecting and attributing AI-generated or edited images, to our setting. In this case, we train an $(n + 1)$-class ResNet-18 classifier using our labeled positive and negative suspicious images as the training data. Finally, we include the two additional variants of our approach described in Section 4.3, denoted as EditTrack-Bin and EditTrack-Bin-Multiple.

**Evaluation metrics:** We report both *detection accuracy* and *attribution accuracy*. For testing positive pairs, detection accuracy is defined as the fraction of pairs in which the suspicious image is correctly identified as edited from the base image, while for testing negative pairs it is defined as the fraction correctly classified as not edited from the base image. Attribution accuracy for testing positive pairs is defined as the fraction of pairs in which the suspicious image is correctly attributed to the editing model that generated it. Note that attribution accuracy for negative pairs is the same as detection accuracy, and thus we omit it for negative pairs. We also report *overall detection/attribution accuracy*, computed as the average accuracy of correctly detecting or attributing each pair across all testing positive and negative datasets.

**Parameter settings:** Unless otherwise specified, we use the default settings and hyperparameters for all image editing models and baseline methods. In our EditTrack, the multi-class classifier is implemented as a three-layer MLP, with the hidden layer dimension set to 30, trained for 1,000 epochs using a learning rate of 0.001, a batch size of 16, and a dropout rate of 0.1.

## 5.2 MAIN RESULTS

Table 2 and 3 report the detection and attribution results of all methods across different datasets. A detailed breakdown of each method's performance across editing models is provided in Tables 5–11 in the Appendix. First, we observe that directly prompting or fine-tuning pre-trained VLMs yields limited performance on the image-editing detection and attribution task. Specifically, QWen-Prompting performs poorly on positive pairs, achieving only about 10% detection accuracy and attribution accuracy close to 0. QWen-FT also fails to accurately detect and attribute positive pairs, even after fine-tuning on our

Table 3: Attribution accuracy results across methods and datasets. "-" indicates this method is not applicable for the attribution task.

| Method | Positive Pairs | | Overall Acc. |
|---|---|---|---|
| | Flickr2K | WikiArt | |
| QWen-Prompting | 0.004 | 0.008 | 0.440 |
| QWen-FT | 0.000 | 0.000 | 0.444 |
| LLaVa-FT | 0.224 | 0.208 | 0.148 |
| WAM | - | - | - |
| GRE | 0.332 | 0.428 | 0.519 |
| EditTrack-Bin | 0.436 | 0.356 | 0.572 |
| EditTrack-Bin-Multiple | 0.864 | 0.812 | 0.903 |
| EditTrack | 0.976 | 0.952 | 0.976 |

training datasets; and LLaVa-FT inaccurately detects and attributes a large portion of negative pairs.

Second, adapting existing AI-generated image detection and attribution methods to our setting also yields limited performance. Specifically, WAM shows partial robustness against several diffusion-

based editing models but is completely ineffective against the autoregressive editing model EditAR, leading to low detection accuracy on positive pairs. GRE exhibits poor performance across both positive and negative pairs because it overlooks the unique characteristics introduced during the editing process as well as the relationship between the suspicious image and its corresponding base image. Moreover, GRE particularly fails on our unrelated dataset. Third, EditTrack-Bin and EditTrack-Bin-Multiple achieve reasonably good detection performance but remain suboptimal since they treat editing models in isolation; when applied to the more challenging attribution task, they become increasingly inaccurate. Overall, we find that EditTrack consistently and substantially outperforms all baselines, achieving high detection and attribution accuracy across both positive and negative pairs.

## 5.3 ABLATION STUDIES

**Impact of re-edited images:** The features used in EditTrack are derived from two groups of re-edited images: re-editing the base images and re-editing the suspicious images. Figure 3a illustrates the contribution of each group by reporting overall detection and attribution accuracy. "Suspicious-only" indicates that only features from re-edited suspicious images are used, "Base-only" indicates that only features from re-edited base images are used, and "Combined" indicates that features from both groups are used, which is the default configuration of EditTrack. The results demonstrate that both sources of features provide complementary information, and leveraging them together yields the best performance for EditTrack.

**Impact of similarity metrics:** Our Edit-Track employs six similarity metrics: Bhattacharyya distance, intersection score, LPIPS, CLIP score, pHash, and structural distance. Figure 3b illustrates the impact of incrementally adding each metric to EditTrack by reporting overall detection and attribution accuracy. The "Number of Used Similarity Metrics" denotes that only the first $k$ metrics are used to train and test the classifier. The results show that each additional metric contributes to improving the performance of EditTrack.

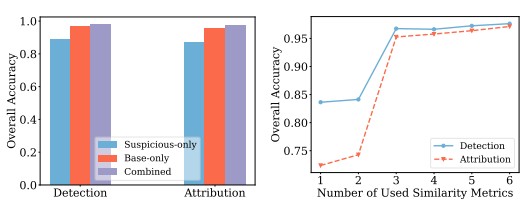

(a) Re-edited Images     (b) Similarity Metrics

Figure 3: Ablation studies.

**Attribution to an unseen editing model:** In this scenario, we evaluate EditTrack when positive pairs can be created by an editing model that was not included during training. Specifically, we consider four candidate editing models as seen during training, while treating the fifth model (Fire-Flow) as unseen; the training dataset excludes positive pairs from FireFlow. Accordingly, we train a 5-class classifier using EditTrack. To enable attribution to an unseen editing model, we introduce a threshold $\tau$ for the 'non-edited' label: given a base–suspicious image pair, if the 'non-edited' label has the highest probability, the pair is classified as 'non-edited' only if this probability exceeds $\tau$; otherwise, it is classified as 'edited by unseen model.' We reserve 20% of the training negative pairs as a validation set and select $\tau$ such that the detection accuracy on these validation negative pairs is 0.9. We then evaluate detection and attribution performance on our test datasets. As shown in Table 4 in the Appendix, even for an unseen model, EditTrack achieves a detection accuracy of 0.91 and an attribution accuracy of 0.83.

## 6 CONCLUSION AND FUTURE WORK

In this work, we propose EditTrack, the first framework for detecting and attributing AI-assisted image editing. Our approach shows that capturing the artifacts introduced during the editing process through a re-editing procedure enables accurate detection of whether a suspicious image is derived from a base image via an editing model, and further allows attribution to the specific model responsible. This re-editing-based method outperforms existing AI-generated image detection and attribution techniques when adapted to this setting. An interesting direction for future work is to extend EditTrack to text and video, investigating the feasibility of detecting and attributing whether a suspicious text or video has been edited from a base version using AI models.

## 7 ETHICS STATEMENT

Our work, EditTrack, aims to address the growing concerns surrounding the misuse of AI-assisted image editing, specifically copyright infringement and the proliferation of deepfakes. While our framework provides a powerful tool for detection and attribution, which can be beneficial for artists and forensic analysts, we acknowledge that its capabilities could also be misused. For instance, it might be adapted to trace the origin of benign edits, potentially infringing on user privacy. We have carefully considered these ethical implications and designed EditTrack to operate without requiring access to the original editing prompts or model parameters, thereby limiting its data collection requirements and protecting user privacy. Our method is a post-hoc mechanism intended for scenarios where edited images have already been produced, providing a way to mitigate harm after the fact rather than preemptively preventing creative use of these tools. We commit to making our work available in a manner that promotes responsible use and discourages malicious applications.

## 8 REPRODUCIBILITY STATEMENT

To ensure reproducibility, we provide detailed descriptions of our methodology, experimental setup, and evaluation. The EditTrack framework is presented in Section 4, covering the four key observations, the re-editing procedure, and the selection of similarity metrics and classifier. Section 5.1 describes the experimental setup, including the five open-source AI editing models, the construction of training and testing datasets, and the hyperparameter settings for our classifier. The evaluation metrics and results across all methods and datasets are reported in Tables 2 and 3, with additional results provided in the Appendix. Our results can be reproduced using this information together with publicly available resources. We will also release our code and datasets.

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

Table 4: Detailed attribution results of EditTrack when testing dataset includes an unseen editing model (FireFlow). Each row shows results for an editing model, which is used for producing the suspicious images in positive pairs. Each column indicates an output class of attribution. "Non-edit" indicates the suspicious image in the pair was not edited from the base image. "Unseen" indicates the suspicious image in the pair was edited from the base image using an unseen model.

| Editing Model | DiffEdit | StableFlow | Step1X-Edit | EditAR | Non-edited | Unseen |
|---|---|---|---|---|---|---|
| DiffEdit | 1.00 | 0 | 0 | 0 | 0 | 0 |
| StableFlow | 0 | 0.90 | 0.08 | 0 | 0 | 0.02 |
| Step1X-Edit | 0.03 | 0 | 0.81 | 0.01 | 0.02 | 0.13 |
| EditAR | 0 | 0 | 0 | 0.98 | 0.01 | 0.01 |
| Negative Pairs | 0 | 0 | 0.0075 | 0 | 0.7825 | 0.21 |
| FireFlow (Unseen) | 0 | 0 | 0.02 | 0.06 | 0.09 | 0.83 |

Table 5: Detailed attribution results of EditTrack. Each cell represents the fraction of positive pairs generated by an editing model or negative pairs (rows) that are attributed to an editing model or classified as non-edited (columns).

| Editing Model | DiffEdit | FireFlow | StableFlow | Step1X-Edit | EditAR | Non-edited |
|---|---|---|---|---|---|---|
| DiffEdit | 1.00 | 0 | 0 | 0 | 0 | 0 |
| FireFlow | 0 | 0.99 | 0 | 0 | 0 | 0.01 |
| StableFlow | 0 | 0.01 | 0.95 | 0.04 | 0.01 | 0 |
| Step1X-Edit | 0 | 0 | 0.01 | 0.88 | 0.01 | 0.1 |
| EditAR | 0 | 0 | 0 | 0 | 1 | 0 |
| Negative Pairs | 0 | 0.0025 | 0 | 0.0075 | 0 | 0.99 |

Table 6: Detailed attribution results of QWen-Prompting.

| Editing Model | DiffEdit | FireFlow | StableFlow | Step1X-Edit | EditAR | Non-edited |
|---|---|---|---|---|---|---|
| DiffEdit | 0.01 | 0.01 | 0 | 0 | 0.16 | 0.82 |
| FireFlow | 0 | 0 | 0 | 0 | 0.01 | 0.99 |
| StableFlow | 0 | 0.01 | 0 | 0 | 0.04 | 0.95 |
| Step1X-Edit | 0 | 0.05 | 0 | 0 | 0.16 | 0.79 |
| EditAR | 0 | 0.01 | 0 | 0 | 0.02 | 0.97 |
| Negative Pairs | 0 | 0.01 | 0 | 0 | 0.0075 | 0.9825 |

## A  USE OF LLMS

We use large language models to aid or polish writing at the sentence level, such as fixing grammar and re-wording sentences. LLMs were not involved in designing methods, conducting experiments, or drawing conclusions. No sensitive or proprietary data were shared with LLMs.

Table 7: Detailed attribution results of QWen-FT.

| Editing Model | DiffEdit | FireFlow | StableFlow | Step1X-Edit | EditAR | Non-edited |
|---|---|---|---|---|---|---|
| DiffEdit | 0.01 | 0.01 | 0 | 0 | 0.16 | 0.82 |
| FireFlow | 0 | 0 | 0 | 0 | 0.01 | 0.99 |
| StableFlow | 0 | 0.01 | 0 | 0 | 0.04 | 0.95 |
| Step1X-Edit | 0 | 0.05 | 0 | 0 | 0.16 | 0.79 |
| EditAR | 0 | 0.01 | 0 | 0 | 0.02 | 0.97 |
| Negative Pairs | 0 | 0.01 | 0 | 0 | 0.0075 | 0.9825 |

Table 8: Detailed attribution results of LLaVa-FT.

| Editing Model | DiffEdit | FireFlow | StableFlow | Step1X-Edit | EditAR | Non-edited |
|---|---|---|---|---|---|---|
| DiffEdit | 0.23 | 0.22 | 0.09 | 0.28 | 0.16 | 0.02 |
| FireFlow | 0.16 | 0.25 | 0.16 | 0.23 | 0.18 | 0.02 |
| StableFlow | 0.21 | 0.16 | 0.16 | 0.28 | 0.16 | 0.03 |
| Step1X-Edit | 0.23 | 0.23 | 0.08 | 0.26 | 0.18 | 0.02 |
| EditAR | 0.2 | 0.17 | 0.21 | 0.22 | 0.18 | 0.02 |
| Negative Pairs | 0.22 | 0.195 | 0.0975 | 0.2475 | 0.1775 | 0.0625 |

Table 9: Detailed attribution results of GRE.

| Editing Model | DiffEdit | FireFlow | StableFlow | Step1X-Edit | EditAR | Non-edited |
|---|---|---|---|---|---|---|
| DiffEdit | 0.54 | 0.02 | 0.03 | 0.04 | 0.02 | 0.35 |
| FireFlow | 0.07 | 0.44 | 0.02 | 0.06 | 0.01 | 0.4 |
| StableFlow | 0.07 | 0.02 | 0.51 | 0.03 | 0.02 | 0.35 |
| Step1X-Edit | 0.13 | 0.14 | 0.07 | 0.12 | 0.07 | 0.47 |
| EditAR | 0.14 | 0.11 | 0.06 | 0 | 0.29 | 0.4 |
| Negative Pairs | 0.06 | 0.1 | 0.085 | 0.03 | 0.0325 | 0.6925 |

Table 10: Detailed attribution results of EditTrack-Bin.

| Editing Model | DiffEdit | FireFlow | StableFlow | Step1X-Edit | EditAR | Non-edited |
|---|---|---|---|---|---|---|
| DiffEdit | 0.45 | 0.28 | 0.05 | 0.21 | 0.01 | 0 |
| FireFlow | 0.29 | 0.40 | 0.01 | 0.18 | 0.12 | 0 |
| StableFlow | 0.51 | 0.22 | 0.08 | 0.16 | 0.03 | 0 |
| Step1X-Edit | 0.32 | 0.22 | 0.02 | 0.36 | 0.05 | 0.03 |
| EditAR | 0.13 | 0.09 | 0.01 | 0.08 | 0.69 | 0 |
| Negative Pairs | 0.045 | 0.07 | 0.0075 | 0.04 | 0.045 | 0.7925 |

Table 11: Detailed attribution results of EditTrack-Bin-Multiple.

| Editing Model | DiffEdit | FireFlow | StableFlow | Step1X-Edit | EditAR | Non-edited |
|---|---|---|---|---|---|---|
| DiffEdit | 0.93 | 0 | 0 | 0 | 0.07 | 0 |
| FireFlow | 0 | 0.92 | 0 | 0.01 | 0.01 | 0.06 |
| StableFlow | 0.01 | 0.07 | 0.72 | 0.05 | 0.14 | 0.01 |
| Step1X-Edit | 0.02 | 0.11 | 0.06 | 0.64 | 0.05 | 0.12 |
| EditAR | 0 | 0 | 0 | 0 | 0.98 | 0.02 |
| Negative Pairs | 0 | 0.0025 | 0.0025 | 0.01 | 0 | 0.985 |

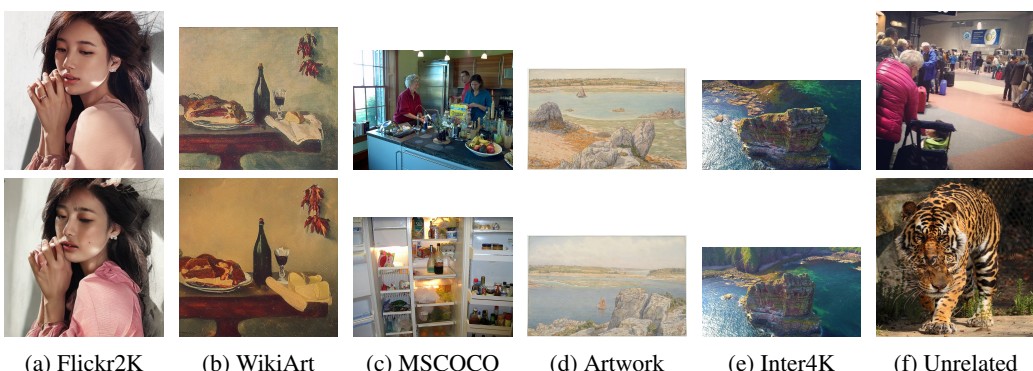

(a) Flickr2K  (b) WikiArt  (c) MSCOCO  (d) Artwork  (e) Inter4K  (f) Unrelated

Figure 4: Image pair samples from different datasets. The first row shows base images, and the second row shows their corresponding suspicious images.

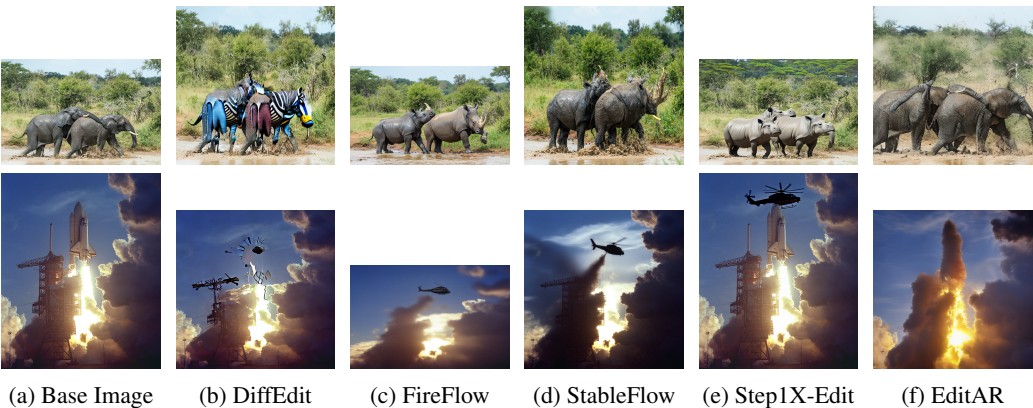

(a) Base Image  (b) DiffEdit  (c) FireFlow  (d) StableFlow  (e) Step1X-Edit  (f) EditAR

Figure 5: Image samples generated using different editing models. The first column shows the base images. The editing prompts are: first row-"Do the image editing task; origin prompt: two **elephants** playfully interact while splashing through a muddy waterhole in a lush, green landscape, editing prompt: two **rhinoceros** playfully interact while splashing through a muddy waterhole in a lush, green landscape"; second row-"Do the image editing task; origin prompt: a **space shuttle** launches dramatically amidst billowing smoke and towering clouds against a clear sky, editing prompt: a **helicopter** rises swiftly amidst swirling dust and towering clouds against a clear sky".

