# OpenReview forum: "EditTrack: Detecting and Attributing AI-assisted Image Editing"
_ICLR.cc/2026/Conference — ICLR 2026 Conference Withdrawn Submission_

### Official Review · Reviewer_kJn8 · 2025-10-28

**Soundness:** 3
**Presentation:** 2
**Contribution:** 2
**Rating:** 2
**Confidence:** 4

**Summary:**

This paper investigates the problem of image edit detection and attribution given a reference image. The overall idea is quite straightforward: the authors re-edit the reference image using candidate models, compare the results with the suspect image, and then train a multi-class classifier to select the editing model that produces the most similar output, or determine that the image was not edited by any known model. For similarity evaluation, the authors consider three perspectives — structural, semantic, and low-level pixel features — and select two metrics for each, forming a 12-dimensional feature vector as input to an MLP. The model design is very simple, with a very small number of parameters.

**Strengths:**

1. The proposed approach is simple, easy to reproduce, lightweight, and resource-efficient.
2. The writing is well-organized and clearly presented, making the paper easy to understand.

**Weaknesses:**

1. The test set is too small. For a classification task, conclusions based on only a few hundred test images lack sufficient reliability and need further validation.
2. The paper lacks comparative experiments. The evaluation should be extended to more Vision-Language Models (VLMs), such as DeepSeek-VL2. Using only Qwen as the sole comparison significantly weakens the persuasiveness of the results.
3. The proposed idea has notable limitations:
   - **Poor scalability:** The method does not generalize well. For instance, expanding from 5 editing models to 8 often requires retraining the entire model from scratch.
   - **Coupled detection and attribution:** The approach integrates detection and attribution into a single process, which can severely affect detection performance when the set of candidate editing models is limited (i.e., when the model overfits to a few specific editing domains). This coupling makes the two tasks interdependent. Although the authors introduce a threshold-based strategy to allow for unseen domains, this empirical thresholding approach remains inherently limited and inflexible.
4. The paper contains too few figures, and the visual presentation is seriously insufficient. More visualizations should be included to better illustrate the dataset characteristics, methodology, and experimental results.

**Questions:**

Please refer to the *Weakness* section for detailed explanations.

---

### Official Review · Reviewer_yyN9 · 2025-10-30

**Soundness:** 2
**Presentation:** 2
**Contribution:** 2
**Rating:** 2
**Confidence:** 4

**Summary:**

This paper introduces EditTrack, a novel framework for detecting and attributing AI-assisted image editing. Unlike existing methods that focus on identifying AI-generated images, EditTrack aims to determine whether a suspicious image is derived from a specific base image using an AI editing model and, if so, which model was responsible. The framework is built on four key observations about AI image editing and uses a re-editing strategy combined with multiple similarity metrics to perform both detection and attribution. Extensive experiments demonstrate that EditTrack outperforms existing baselines across multiple datasets and editing models, offering a practical tool for applications such as copyright protection and forensic analysis.

**Strengths:**

1、The paper tackles a pressing challenge in AI image editing, focusing on the detection and attribution of edits, which is critical for addressing deepfake issues and copyright concerns in the digital age.

2、EditTrack introduces a novel re-editing strategy based on four key observations about AI editing models, offering a more effective approach to detecting and attributing image edits compared to traditional methods.

3、The method goes beyond detecting AI-generated content by identifying the specific editing model used, filling a significant gap in current AI detection frameworks.

**Weaknesses:**

1、High Computational Cost and Complexity: EditTrack requires a computationally expensive process of re-editing images multiple times with each candidate editing model and comparing the generated results. This process demands significant computational resources, which could be problematic when dealing with large datasets or complex images.

2、Dependence on Known Editing Models: EditTrack relies on predefined candidate editing models, limiting its ability to detect and attribute edits from models that are not included in the candidate set. This dependency restricts its flexibility, especially when new or emerging editing models are used.
Furthermore, it heavily relies on multiple similarity metrics (structural, semantic, and pixel-level) to compare re-edited images. This reliance can limit its effectiveness, particularly for complex or irregular image edits that may not align well with the predefined metrics.

3、Limited Selection of Editing Models: The choice of editing models evaluated in this work primarily focuses on methods with relatively lower performance, and does not incorporate some of the more advanced and widely used editing models such as GPT-Image-1, Qwen-Image, and Nano-Banana. Including these more state-of-the-art models could provide a more comprehensive evaluation and better reflect the current landscape of powerful editing tools.

4、Insufficient Dataset Size and Homogeneity: The dataset used in this study is relatively small, both in terms of the training and testing sets. Additionally, the use of data from the same domain for both training and testing could result in performance metrics that are very close to overfitting, with accuracy nearing 100%. Expanding the dataset and testing across more diverse domains would help to mitigate this overfitting risk and provide a more generalized evaluation of the proposed method's effectiveness.

5、Unbalanced and Potentially Unfair Comparisons: The comparison methods used in this study may not be entirely fair. For instance, the training of Qwen-2.5-VL seems insufficient, possibly due to overfitting caused by an excessive number of negative pairs. This could explain why Qwen-2.5-VL performs notably worse than LLaVa-v1.5-7B. Additionally, the relatively small and non-representative training and testing datasets may have contributed to this discrepancy. Moreover, while QWen-Prompting was compared, it would have been beneficial to include comparisons with more recent and powerful closed-source models, such as GPT-4o-Prompting and Gemini-2.5-pro-Prompting, especially given the limited dataset size. The current selection of models, primarily from a year ago, may not fully reflect the state-of-the-art capabilities of today's cutting-edge models.

**Questions:**

The quality of the paper's writing and overall presentation could be improved. Not only is the writing unclear in places, but the supporting visualizations and figures are also not sufficiently clear. The method descriptions and result presentations lack the clarity needed to fully convey the contributions of the work, making it difficult for readers to easily follow the methodology and interpret the results. Improving the writing and enhancing the clarity of the visual aids would significantly strengthen the paper's impact.

---

### Official Review · Reviewer_wnmv · 2025-10-31

**Soundness:** 3
**Presentation:** 2
**Contribution:** 2
**Rating:** 6
**Confidence:** 3

**Summary:**

The paper introduces EditTrack, which formally tackles image-editing detection and attribution: given a base image and a suspicious image, it determines whether the latter was derived from the former using an AI editor and, if so, identifies the specific editing model. EditTrack significantly outperforms five baselines across five editing models and six datasets.

**Strengths:**

1. This paper addresses the problem of recognizing and attributing image-editing data. It is a novel setup for models to attribute an edited image to a particular base image and generation models.
2. The writing is very clear and easy to follow.
3. The results are good and analysis in Section 4.2 further backups their design and claim.

**Weaknesses:**

1. I wonder how propriety models, like gpt-4o or gemini, perform under the setups. If EditTrack can reach similar performance, I believe this could further provde the effectiveness of the methods.
2. Can you elaborate more on why "structural, semantic, and pixel-level similarity" are specifically chosen for categorizing similarity? What are your insights on these selections?

**Questions:**

See weakness.

---

### Official Review · Reviewer_xXVn · 2025-10-31

**Soundness:** 2
**Presentation:** 1
**Contribution:** 2
**Rating:** 2
**Confidence:** 3

**Summary:**

The paper introduces EditTrack, a framework for detecting whether a suspicious image was AI-edited from a base image and attributing the edit to specific models. The approach employs a re-editing strategy combined with similarity metrics across structural, semantic, and pixel-level dimensions, feeding these features into an (n+1)-class classifier. While the problem formulation is novel and the experimental results demonstrate improvement over adapted baselines, the paper exhibits significant methodological weaknesses, inadequate theoretical grounding, and limited experimental scope.

**Strengths:**

* The paired base-suspicious image detection task is well-motivated and addresses a practical need in content authenticity verification. The distinction from general AI-generated image detection is clearly articulated.
* The multi-faceted similarity measurement across structural, semantic, and pixel-level dimensions provides richer features than single-metric approaches. The ablation study (Figure 3b) demonstrates that each metric contributes to performance.
* Within the limited experimental scope, EditTrack substantially outperforms adapted baselines (Tables 2-3). The overall detection accuracy of 98.3% and attribution accuracy of 97.6% on tested positive pairs demonstrate the method's potential effectiveness.

**Weaknesses:**

* The related work section suffers from substantial gaps and repetitive framing that fail to position the contribution properly. The paper repeatedly states that existing methods "cannot establish whether an image was derived from a given base image" without offering a substantive analysis of specific approaches

* The four key observations (Robustness, Stability, Variety, Dissimilarity) are presented as empirical findings validated on merely 50 positive pairs with only two models (Step1X-Edit and EditAR), which provides insufficient justification for foundational claims. The paper uses ambiguous notation (≈, ≉) that lacks a formal mathematical definition. No threshold or quantitative criterion specifies when similarity holds. Figure 2 displays distribution overlap, but this does not prove the observations hold universally; for extreme edits, high semantic changes, or certain architecture types, these assumptions may fail. Under what specific conditions do these observations hold or break down? What theoretical guarantees can be provided? The authors should provide formal definitions, derive theoretical conditions for validity, and conduct failure mode analysis rather than relying on limited empirical observation.

* The method critically depends on generating an accurate proxy prompt $p^{′}_e$ via BLIP-2 captioning. What happens when captions are poor, misleading, or domain-specific? No analysis provided. The paper does not investigate robustness to caption models (CLIP, LLaVa as alternatives) or the semantic gap between the true editing prompt pe and proxy $p^{′}_e$. Since editing prompts can be highly specific ("Inpaint the cat's eyes to be glowing neon blue"), the difference between vague captions and precise instructions could be substantial. Does caption length, specificity level, or semantic accuracy correlate with detection performance? The assumption that "semantically similar prompts" (pe ≈ $p^{′}_e$) lead to similar outputs needs empirical validation across the five models. Additionally, different editing models may be robust to prompt variations differently—have model-specific calibrations been explored?

* The approach concatenates 12n features from six metrics without principled justification: why these specific metrics rather than traditional forensic features (frequency analysis, noise patterns, texture inconsistencies)? The ablation (Figure 3b) shows incremental improvement with each metric, but this does not validate the choice of metrics themselves or their orthogonality. How would a simple nearest-neighbor baseline perform using the same six similarity metrics? What about standard forensic classifiers (SVM, Random Forest) on the feature vectors? The baseline section acknowledges EditTrack-Bin variants achieve 0.904-0.970 overall accuracy (Table 2), which raises the question: how much improvement comes from the (n+1)-class classifier versus the features themselves? The classifier itself, a simple 3-layer MLP with 30 hidden units appears arbitrary without justification or hyperparameter ablation. Have other architectures (Transformers, kernel methods) been explored?

* WAM watermarking is fundamentally incompatible with editing (acknowledged in the paper), making it a strawman; GRE was designed for generation attribution, not paired detection; VLM baselines (QWen, LLaVa) are prompted without task-appropriate fine-tuning architecture. Critically missing comparisons to forensic methods specifically designed for edit detection and localization (e.g., FakeShield, Noise-assisted Prompt Learning for Forgery Detection, EditScout, DiffForensic). The paper should adapt these methods properly or explain why they are inapplicable.

* The paper omits recent powerful editors like Emu Edit, UltraEdit, FoundHand and other , ControlNet-based editors.

**Questions:**

* The four foundational observations are validated on only 50 image pairs with two models. Can you provide formal definitions of similarity thresholds and theoretical conditions for validity? What is the failure mode analysis across different architectural types?

* Proxy prompt generation depends entirely on BLIP-2 caption quality with no robustness evaluation. How does caption quality affect detection performance? What is the semantic gap between true editing prompts and proxies?

* Evaluation covers only five models, missing recent SOTA (Emu Edit, UltraEdit, ControlNet, domain-specific editors). How does EditTrack generalize across different architectures? Why does performance drop significantly on unseen models (98.3% → 91%)?

* The six-metric feature concatenation lacks justification compared to simpler alternatives. How much improvement comes from features vs. classifier? How would nearest-neighbor or SVM perform on identical features?

* Baselines are poorly adapted (WAM, GRE incompatible with task). Why not compare against forensic methods designed for edit detection (FakeShield, EditScout, DiffForensics)?

* No robustness evaluation against post-processing (JPEG, rotation), adversarial perturbations, or counter-forensic attacks. How does performance generalize across different image distributions?

---

### Note · Authors · 2025-11-12

I have read and agree with the venue's withdrawal policy on behalf of myself and my co-authors.